# Mechanisms of Resistance to Anti-CD38 Daratumumab in Multiple Myeloma

**DOI:** 10.3390/cells9010167

**Published:** 2020-01-09

**Authors:** Ilaria Saltarella, Vanessa Desantis, Assunta Melaccio, Antonio Giovanni Solimando, Aurelia Lamanuzzi, Roberto Ria, Clelia Tiziana Storlazzi, Maria Addolorata Mariggiò, Angelo Vacca, Maria Antonia Frassanito

**Affiliations:** 1Department of Biomedical Sciences and Human Oncology, Unit of Internal Medicine and Clinical Oncology, University of Bari “Aldo Moro”, 70124 Bari, Italy; ilaria.saltarella@libero.it (I.S.); vanessa86.desantis@gmail.com (V.D.); assuntamel@hotmail.it (A.M.); antoniogiovannisolimando@gmail.com (A.G.S.); aurelia.lamanuzzi@libero.it (A.L.); roberto.ria@uniba.it (R.R.); 2Department of Biology, University of Bari “Aldo Moro”, 70125 Bari, Italy; tizianastorlazzi@gmail.com; 3Department of Biomedical Sciences and Human Oncology, Unit of General Pathology, University of Bari “Aldo Moro”, 70124 Bari, Italy; mariaaddolorata.mariggio@uniba.it (M.A.M.); antofrassanito@gmail.com (M.A.F.)

**Keywords:** multiple myeloma, CD38 antigen, daratumumab, drug resistance

## Abstract

Daratumumab (Dara) is the first-in-class human-specific anti-CD38 mAb approved for the treatment of multiple myeloma (MM). Although recent data have demonstrated very promising results in clinical practice and trials, some patients do not achieve a partial response, and ultimately all patients undergo progression. Dara exerts anti-MM activity via antibody-dependent cell-mediated cytotoxicity (ADCC), antibody-dependent cellular phagocytosis (ADCP), complement-dependent cytotoxicity (CDC), and immunomodulatory effects. Deregulation of these pleiotropic mechanisms may cause development of Dara resistance. Knowledge of this resistance may improve the therapeutic management of MM patients.

## 1. Introduction

Multiple myeloma (MM) is a hematological malignancy characterized by the expansion of malignant plasma cells (MM cells) in the bone marrow (BM). Despite the availability of new anti-MM agents, i.e., proteasome inhibitors (PI) and immunomodulatory drugs (IMiD), MM remains an incurable disease [1,2]. A real-world retrospective study demonstrated that PI and IMiD refractory patients show a median overall survival of approximately eight months [3].

MM drug resistance closely relies on tumor microenvironment [4]. MM cells home to and expand in the BM, establishing a relationship with BM stromal cells (SCs) (i.e., endothelial cells, fibroblasts, osteoblasts, osteoclasts, and immune cells) [5,6,7,8], thus creating a supportive niche. Inside the niche, the cellular/noncellular components favor MM cell survival through the activation of several biological processes, i.e., angiogenesis, hypoxia, autophagy, metabolism reprogramming, and apoptosis resistance, which gradually modify tumor microenvironment and BMSCs [9,10,11,12,13]. Accordingly, targeting both tumor and nontumor cells is the main goal of the new anti-MM therapeutic approaches.

CD38, a transmembrane glycoprotein, was initially identified as a marker on activated T cells [14,15], nowadays defined as a cell receptor and a cell-surface enzyme [16,17]. It is widely expressed on lymphoid (i.e., T and B lymphocytes, NK cells, and monocytes) and non-lymphoid tissues. Notably, it is highly expressed in MM cells and generally used as a co-marker for their identification and enumeration.

CD38 contributes to the catabolism of nicotinamide adenine dinucleotide (NAD^+^) involved in several cellular functions, including local immunosuppression [13]. In the BM niche, MM cells reprogram their metabolism that increases NAD^+^ levels yielding lactate and H^+^ [18]. Through its ectoenzymatic activity, CD38 catalyzes the cleavage of NAD^+^ into cyclic adenosine diphosphate (ADP)-ribose, next to ADP-ribose via hydrolase activity [13]. Furthermore, CD38 catalyzes the transglycosilation of nicotinamide adenine dinucleotide phosphate (NADP) and nicotinic acid, to yield nicotinic acid adenine dinucleotide phosphate (NAADP). Cyclic ADP-ribose and NAADP are second-messenger molecules which trigger calcium release from inositol trisphosphate (IP3)—independent intracellular stores. ADP-ribose is the substrate for the formation of adenosine (ADO) that is involved in the regulation of MM immune autocrine and paracrine signals [18].

Daratumumab (Dara) is the first-in-class human-specific anti-CD38 IgG1 mAb approved for the treatment of MM [19]. Recent data demonstrated encouraging response rates in clinical trials, both as a single agent and in combination regimens in the relapsed/refractory setting [19,20].

Dara activity as a single agent has been demonstrated in the GEN501 and Sirius clinical trials [21,22] that show a 31.1% overall response rate (ORR), a 20.1-month overall survival (OS), and a four-month progression-free survival (PFS) in relapsed/refractory patients [21].

The deep response and prolonged PFS can be achieved in MM patients by combining Dara with IMiD and PI regimens. In phase-three trials (POLLUX and CASTOR), the addition of Dara to standard-of-care regimens proved a higher response rate, depth of response, and PFS in patients with >1 prior lines of therapy [23,24]. Due to these results, Dara is currently approved by the European Medicines Agency (EMA) and also by the US Food and Drug Administration (US FDA) as a single agent in relapsed/refractory MMand in combination with either Dara-Revlimid-Dexamethasone (DRd) and Dara-Velcade-Dexamethasone (DVd) in patients with at least one previous line of therapy. Dara in newly diagnosed transplant ineligible patients was investigated in a recent phase III study [25] that compared the gold-standard Bortezomib-Melphalan-Prednisone (VMP) regimen to VMP plus Dara. Results showed significantly increased ORR rates and improved complete response and PFS in Dara-treated patients [25]. Therefore, FDA- and EMA-approved Dara plus VMP for newly diagnosed patients not eligible for autologous stem-cell transplantation. In transplant-eligible patients with newly diagnosed MM, CASSIOPEA study [26] demonstrated that the association of Dara with standard of care Velcade-Thalidomide-Dexamethasone (VTD) improves depth of response and PFS, suggesting a concrete change in the scenario of this stage of disease.

However, despite the well-established clinical efficacy of Dara, approximately 60% of patients do not achieve a partial response, and ultimately all patients undergo progression [27]. In this review, we will describe the mechanisms involved in Dara efficacy and resistance, focusing on tumor cells and BM microenvironment.

## 2. Mechanisms of Action

Dara exerts anti-MM activity via several Fc-dependent mechanisms, namely the antibody dependent cell-mediated cytotoxicity (ADCC), the antibody-dependent cellular phagocytosis (ADCP), and the complement-dependent cytotoxicity (CDC). Furthermore, Dara plays an immunomodulatory effect via withdrawal of the CD38-expressing immune suppressor cells. Pleiotropic mechanisms of action are illustrated in Figure 1.

### 2.1. ADCC

ADCC is the most important antibody-mediated mechanism of target cell killing. The binding of Fcγ receptors (FcγRs) expressed on effector cells to the Fc fragment of the CD38-targeting antibodies induces the intracellular phosphorylation of the immune tyrosine-based activating motifs of the FcγRs that results in the lysis of target cells [28]. The main FcγRs expressed on effector cells are CD32c and CD16 on NK cells and CD64 on macrophages, dendritic cells, neutrophils, and eosinophils [29,30]. NK cells kill target cells via degranulation and release of perforins and granzymes [31] and via activation of receptors cell death (i.e., the family of tumor necrosis factor receptors, TNFRs), that induce target cells apoptosis [32] (Figure 1). The engagement of CD16 triggers the secretion of pro-inflammatory cytokines, such as interferon gamma (INFγ), TNFα, macrophage colony-stimulating factor (M-CSF), monocyte chemoattractant protein-1 (MCP-1), chemokine ligand (CCL)-3, and CCL-4, CCL-5, that contribute to the recruitment of adaptive immune cells [33]. The different activity of NK cells has been associated with two distinct cell subsets, based on the expression of CD16 and CD56. Indeed, CD56^dim^CD16^+^ NK cells that prevail in peripheral blood (PB) have a potent cytolytic activity and release high amounts of INFγ early after activation [34]. On the contrary, the CD56^bright^CD16^+^/^−^ NK cell population that locates in tissues and in secondary lymphoid organs is mainly involved in the late cytokine production. These data suggest an important coordination between the two different cell subsets in sustaining the killing of target cells and in recruiting other immune cells to promote the adaptive response. Research by de Weers et al. [35] demonstrated that Dara is able to induce a dose-dependent killing of MM cells via ADCC, in both sensitive and drug-resistant patients. Interestingly, Dara maintains its cytotoxic activity also in the presence of BMSCs, although a decrease in the killing activity is due to the protective role of BMSCs for MM cells survival.

Several studies documented that therapeutic mAbs sustain ADCC by other immune cells (Figure 1) [28,36,37]. Neutrophils exert cytotoxic activity through the destruction of plasma membrane, leading to the lysis of target cells [36]. Although macrophages are mainly involved in ADCP (see below), M1 macrophages induce ADCC via the release of nitric oxide (NO) and other oxygen reactive species (ROS) [28]. Finally, γδ T cells contribute to target cell killing via ADCC through a CD16-dependent mechanism that induces γδ T cells’ degranulation and release of the lysosomal-associated membrane protein-1 (LAMP-1 or CD107a) [37]. However, the role of neutrophils and γδ T cells in Dara-induced ADCC is currently unknown.

### 2.2. ADCP

ADCP is an Fc-dependent mechanism that supports antibody-mediated clearance of tumor cells both in vitro and in vivo [38]. Specifically, antibody-opsonized target cells activate the FcγRs expressed on the surface of different immune cells that promote the phagocytic process through the internalization and the consequent lysis of target cancer cells via phagosome acidification [38].

Overdijk et al. [39] studied the ability of Dara to induce tumor-associated macrophage (TAM)-mediated phagocytosis in MM (Figure 1). Flow cytometry and live-cell imaging revealed that macrophages rapidly engulf CD38^+^ MM cell lines in vitro. Authors finely demonstrated that phagocytosis contributes to the antitumor activity of Dara in vivo by using immune-deficient SCID-BEIGE mice lacking B, T, and NK cells [39]. Dara-dependent phagocytosis is related to CD38 expression, in that MM cell lines with low CD38 levels are not susceptible to Dara effect. They also explored the role of Dara-dependent phagocytosis in fresh purified MM cells derived from MM patients, suggesting that phagocytosis is a potent mechanism that contributes to the therapeutic activity and clinical efficacy of Dara in MM patients [39].

### 2.3. CDC

The engagement of the Fc fragment to C1q initiates the classical pathway of complement cascade, leading to formation of the membrane attack complex (MAC), a transmembrane channel that enables the osmotic lysis of the target cell (Figure 1) [40]. The activation of the complement cascade induces the generation of the active opsonin C3b that covalently binds to glycoproteins on the target cell surface and allows cell recognition by phagocytic cells and production of the anaphylatoxins C3a and C5a [40]. These pro-inflammatory chemokines modulate several immune responses promoting leukocyte chemotaxis, phagocytes degranulation and the simultaneous upregulation of activating FcγRIII and downregulation of inhibitory FcγRs on macrophages [41].

Dara is the most important inducer of CDC. It was selected from a panel of 42 mAbs, based on its ability to produce the complement-mediated lysis of the B lymphoblast Daudi cell line and of freshly isolated MM cells alone and in co-culture with BMSCs [35]. Dara induces CDC at low concentration of human serum, proving its efficacy in a complement-defective milieu as the BM of MM patients [35]. In addition, Nijhof et al. [42] found that Dara-mediated CDC depends on CD38 expression. Indeed, enforced CD38 expression on MM cells significantly improved CDC activity of Dara [42].

### 2.4. Immuno-Modulatory Effects

Dara treatment modulates the enzymatic activity of CD38 by reducing the ADO levels. The production of ADO in the MM microenvironment is regulated by the “canonical” adenosinergic pathway catalyzed by CD39/CD73, and the “alternative” axis catalyzed by CD38/CD203a/CD73 [43,44]. CD38 starts the “alternative” axis converting NAD^+^ to cyclic ADP-ribose, further metabolized by CD203a into adenosine monophosphate (AMP), which is subsequently converted to ADO by CD73 [18,44]. Using a recombinant CD38 protein, van de Donk et al. [45] showed that Dara modulates the CD38 enzymatic activity in that it reduces cyclase activity and induces hydrolase activity, thus increasing NAD^+^ and ADPR levels and decreasing cADPR (Figure 1) [45].

As CD38 is expressed on several immune cells [46,47,48], Dara treatment depletes CD38^+^ immune cells, causing a modification of the antitumor response. Krejcik et al. [49] showed that Dara treatment reduces the immunosuppressive cells in MM microenvironment, including myeloid-derived suppressor cells (MDSCs), Treg, and Breg cells, and increases the anti-MM activity. Dara treatment depleted CD19^+^C24^+^CD38^+^ Bregs in MM patients immediately after the first infusion and during the entire regimen, as well as when the Dara-treated patients relapsed. Dara effect on MDSCs was investigated by using MM-cell-induced MDSCs generated in vitro from co-cultures of healthy donor PB mononuclear cells (MCs) with MM cell lines [49]. MM-cell-induced MDSCs (CD11b^+^CD14^-^HLA-DR^-^CD33^+^CD15^+^CD38^+^ cells) were strongly reduced after Dara treatment, suggesting their sensitivity to Dara-mediated ADCC and CDC [49].

Krejcik et al. [49] demonstrated the existence of a CD38^+^ Tregs with an increased immune-suppressive activity, compared to the CD38^-/dim^ Tregs. Interestingly, Dara also affects CD38^+^ Tregs, contributing to the reduction of the immune-suppression [49].

Dara induces proliferation of T helper and cytotoxic cells from PB and BM samples, increasing the CD8^+^/CD4^+^ and CD8^+^/Tregs ratios and stimulating T-cell function and activity via the release of INFγ [49]. Also, Dara treatment favors the development of memory T cells and decreases naïve T cells, suggesting a shift from inactive to effector T cells with an immunological memory: these cells can be reactive against tumor antigens (Figure 1) [49].

Chen et al. [50] demonstrated that overexpression of CD38 is a mechanism of tumor escape from PD1/PDL1 axis blockade in that it suppressed CD8^+^ T cell function via ADO signaling. Consistent with these results, Bezman et al. [51] showed an enhanced antitumor activity in MC38 and J558 mice treated with a combined therapy, suggesting that a dual targeting of CD38 and PD1 may represent a promising anti-MM therapeutic strategy. The synergic effect induced by anti-CD38 mAbs was related to a reduction in the frequency of immunosuppressive Tregs and MDSCs populations.

All these data support the immune-modulatory role of Dara that may contribute to its anti-MM activity.

## 3. Mechanisms of Resistance

Although recent data have demonstrated very promising results in Dara clinical practice and trials, some patients may do get a complete response or may experience a relapse. Several mechanisms contribute to the development of Dara resistance, including CD38 reduction, ADCC, ADCP, CDC, and immune-mediated processes, as illustrated in Figure 2.

### 3.1. CD38 Reduction

As Dara recognizes CD38 on MM cells, its antitumoral activity closely depends on CD38 expression. Low-baseline CD38 levels on MM cells were shown in non-responder patients [52], implying that CD38 expression may be a predictor factor of clinical response.

Dara therapy reduces the CD38 expression on MM cells that endures throughout the entire drug regimen. Flow cytometry analysis of CD38 expression on MM cells from patients’ BM samples in the GEN501 study [52] revealed a significant decrease of CD38 expression during the Dara treatment compared to baseline values. Low levels of CD38 expression were also observed during MM progression [52]. Interestingly, CD38 reduction occurs in non-responder and partial-responder patients, as well as in responder ones [52]. Nijhof et al. [52] demonstrated that the low CD38 levels are only a transient effect due to the treatment, in that the CD38 expression is restored 3–6 months after the last drug infusion. Analysis of CD38 expression of MM cells from PB overlaps that of MM cells from BM in terms of CD38^dim/-^ expression and rapid decrease of circulating tumor cells after the first Dara infusion [52]. Lenalidomide and pomalidomide are able to increase CD38 expression on MM cells and to synergize with Dara in vitro and in vivo [53,54,55]. Nevertheless, Krejcik et al. [56] showed a significant decrease of CD38 expression in MM cells of patients enrolled in the GEN503 clinical study, where Dara was combined with lenalidomide and dexamethasone.

Several mechanisms may contribute to the CD38 reduction. The first reported one involves clone selection: Dara selectively depletes the CD38 highly expressing MM cells via ADCC and CDC, that allows expansion of MM cells with low CD38 (Figure 2) [57]. Thus, CD38 expression conveys a higher susceptibility to Dara-mediated anti-MM activity. Upon Dara binding, CD38 rapidly undergoes endocytosis through an internalization process independent of signal transduction pathways, in that both agonistic and non-agonistic mAbs are able to prompt CD38 internalization [58,59]. Horenstein et al. [60] demonstrated that Dara binding to CD38 modifies cytoskeleton reorganization in MM cells by inducing a redistribution of CD38 antigens into polar aggregates released in the BM microenvironment as microvesicles that hence express CD38 and other ectoenzymes (CD39, CD203a, CD73) [11,61]. These microvesicles are able to produce ADO from adenosine triphosphate (ATP) and NAD^+^, thus contributing to the increase of ADO levels in the BM niche, and to generate immune suppression via the modulation of pro- and anti-inflammatory cytokine release. Microvesicles released following Dara treatment may act both locally within the BM and at distance via blood stream, and may be considered one mechanism of Dara resistance that abrogates anti-MM immune responses (Figure 2) [11,62].

Dara-induced CD38 depletion involves not only MM cells but also CD38^+^ immune cells, including NK, B, and T cells. Expression of CD38 is not affected in monocytes, probably due to their role in the uptake of CD38-Dara complexes from MM cells. Krejcik et al. [56] observed that monocytes and granulocytes contribute to the decrease of CD38 by engulfment of CD38-Dara complexes, in the absence of the MM cells phagocytosis, through a process known as trogocytosis. The uptake of CD38-Dara complexes results in the transfer of both Dara and CD38 from MM cells to monocyte and granulocyte surface [56]. In the trogocytosis, the mAb-opsonized target cells (donor cells) bind to the FcγRs of monocytes and granulocytes (acceptor cells), and create an immunologic synapse [63,64,65] that induces the engulfment of CD38, along with the nearby fragment of plasma membrane (Figure 2) [65]. Thus, trogocytosis contributes to CD38 reduction, as well as to the decrease of other surface proteins located nearby the CD38 antigen, including CD49f, CD56, CD54, and CD44, suggesting a potential role of this mechanism in the Dara-mediated downregulation of membrane-associated complement-inhibitory proteins. Following trogocytosis MM cells may display a different surface antigen profile [56].

The use of high mAb doses strongly activates the complement and may exhaust and saturate cell effector functions. These exhausting conditions promote trogocytosis, rather than phagocytosis, suggesting that the excess of mAbs-target antigen complexes may promote the escape of MM cells through the trogocytosis of immune complexes [65].

### 3.2. ADCC Resistance

Activation of ADCC requires the engagement of FcγRs expressed on effector cells. Different variants of FcγRs are related to a different cell response: FcγRIIA-131H and FcγRIIA-158V are associated with a higher affinity to the Fc fragment of mAbs and, thus, to a higher ADCC activity; by contrast, FcγRIIb has a negative immune-modulatory activity. Consequently, the FcγRs variants expression on NK cells may modulate the Dara-mediated antitumor activity via ADCC [57].

NK cells express the CD38 antigen. Dara treatment induces a depletion of PB and BM NK cells via fratricide ADCC against nearby NK cells (Figure 2). Analysis of NK cells of patients in GEN501 and SIRIUS studies showed that NK cell levels decreased immediately after the drug’s first infusion [52]. Dara-mediated NK cells fratricide may strongly affect the NK-mediated ADCC that influences the Dara efficacy and increases the risk of relapse [66]. However, ex vivo experiments showed that the remaining NK cells are still active and able to exert cytotoxic activity [66]. Wang et al. [67] demonstrated that the remaining CD38^−/low^ NK cells, resistant to Dara-induced NK-cell fratricide, have enhanced proliferative and anti-MM activity in vitro and in vivo, with the likelihood to be expanded in vitro. Based on these results, the authors suggested that reinfusion of ex vivo expanded autologous NK cells may improve Dara efficacy [67].

Finally, de Haart et al. [68] showed that BMSCs support resistance to Dara therapy, preventing Dara-mediated ADCC. The microenvironment-related resistance is mediated by the overexpression of the anti-apoptotic protein survivin in MM cells upon interaction with BMSCs. Indeed, treatment of MM cells/BMSCs co-cultures with YM155, a small molecule that inhibits survivin expression, increased Dara-mediated ADCC abrogating the protective role of BM microenvironment against Dara treatment [68].

### 3.3. ADCP Resistance

Another important mechanism of Dara resistance involves CD47, also known as integrin-associated protein (IAP), an antigen identified in different tumors, including MM [69]. CD47 is expressed at low levels on normal plasma cells, while it is overexpressed on MM cells [70]. CD47 overexpression may contribute to the immune escape of tumor cells through the inhibition of ADCP, via the binding of CD47 to the signal-regulatory protein alpha (SIRPα) on TAMs [71]. The CD47/SIRPα complex acts as a “don’t eat me” signal that induces SIRPα phosphorylation and association to Src-homology phosphatase 1 domain (SHP-1) on macrophages, resulting in the inhibition of phagocytosis (Figure 2) [72]. The “don’t eat me” signal is the focus for different mAb therapies able to block CD47 [72,73]. Indeed, van Bommel et al. [73] demonstrated an increase of ADCP through CD47-blocking mAbs and the consequent regulation of the anti-phagocytic activity. Specifically, they described the RTX-CD47 bi-specific antibody that recognizes CD47 single-chain fragment in the antibody variable regions (scFv) and in tandem CD20-targeting scFv derived from the Rituximab association [73]. RTX-CD47 promotes the inactivation of CD47/SIRPα signal and causes the selective removal of CD47^+^CD20^+^ cells through phagocytosis, overall suggesting that the block of CD47 may act as a pro-phagocytic therapeutic approach to enhance the tumoricidal activity of anticancer mAbs, including Dara [73].

### 3.4. CDC Resistance

Host cells are protected from complement activation by fluid phase regulators and/or by the expression of membrane-associated complement-inhibitory proteins, such as CD46 and the glycosyl-phospatidylinositol-anchored proteins, such as CD55 and CD59 [74]. These mechanisms play a crucial role in the immunological escape of tumor cells and in the development of drug resistance [75,76,77]. Nijhof et al. [52] demonstrated that Dara has a great efficacy toward MM and lymphoma cell lines with lower CD59 and CD55 levels. Accordingly, treatment with phospholipase-C, which cleaves CD55 and CD59, increases susceptibility of MM cells to Dara. As in the BM microenvironment tumor cells express heterogeneous levels of membrane-associated complement-inhibitory proteins, Dara therapy invariably depletes CD55^dim^ and CD59^dim^ tumor cells (Figure 2). Accordingly, analysis of MM patients enrolled in the GEN501 study showed an increased expression of CD55 and CD59 on MM cells during disease progression, suggesting that their overexpression is associated to the MM resistant phenotype [52].

### 3.5. The Immune-Mediated Resistance

Recently, some authors [49,78,79] have hypothesized that the immune system may play a key role in the development of Dara resistance.

Krejick et al. [49] and Neri et al. [78] documented low numbers of effector memory T cells in relapsed MM patients. Single cell resolution analysis of BM mononuclear cells (MCs) depleted of CD138^+^ cells from Dara/IMiD-treated patients showed a decreased expression of the co-stimulatory antigen CD28 on T cells and a reduced number of M1 macrophages in resistant and/or progressing patients compared to responder ones [78]. Finally, RNA-sequencing of BMSCs cells depleted of CD138^+^ MM cells showed a different gene expression profile between progressed and Dara naive patients. The Database for Annotation, Visualization, and Integrated Discovery (DAVID) analysis provides the functional interpretation of the downregulated genes, such as tool-like receptor (TLR) 8, CD47, chemokine (C-X-C motif) ligand (CXCL) 10, and CXCL4, highlighting their involvement in the Dara-mediated immune response (Figure 2) [79].

Nevertheless, this intriguing hypothesis should be further investigated.

## 4. New Strategies to Overcome Dara Resistance

Several studies are focusing on the development of new strategies to overcome Dara resistance, such as new optimized drugs and/or combined therapeutic regimens.

The combined regimen Dara–IMiD in clinical trials is offering promising outcomes [22,23] and suggests a synergistic effect, probably due to IMiD’ immunomodulatory effects, which ultimately lead to upregulation of CD38 expression [53,54,55].

A new promising molecule to overcome Dara resistance is ATRA, the active metabolite of vitamin A. CD38 expression is positively regulated by retinoic acids that bind to retinoic acids responsive elements (RARE), located in the promoter and within intron 1 of the *CD38* gene. A new synthetic derivative of ATRA, tamibarotene (a.k.a. Am80), upregulates CD38 expression on MM cells, without the side effect of ATRA, suggesting its clinical potential. Chillemi et al. [58] demonstrated that in vitro treatment of fresh isolated MM cells and MM cell lines with ATRA or tamibarotene enhances CD38 expression. Recently, Nijhof et al. [52] showed that ATRA simultaneously increases CD38 expression and reduces CD55 and CD59 expression in resistant MM cells. Thus, ATRA enhances both Dara-mediated ADCC and CDC activity and may help to prevent Dara resistance mediated by CD38 reduction and membrane-associated complement-inhibitory proteins expression.

Another strategy to overcome resistance to Dara may be the use of other anti-CD38 antibodies with a different mechanism of action, namely Isatuximab (SAR650984), MOR202, and TAK-079 [80]. Isatuximab mediates a direct cytotoxicity against MM cells, in addition to the canonical Fc-dependent mechanisms of action [81]. Indeed, Jiang et al. [81] demonstrated that it induces a CD38-dependent depletion of MM cells via homotypic aggregation-associated cell death by actin cytoskeleton polymerization, caspase-dependent apoptosis, and lysosomal cell death [81]. Furthermore, Isatuximab induces an allosteric modulation of CD38 that results in a higher inhibition of its ecto-enzymatic activity [82]. Clinical trials demonstrated that Isatuximab has a great antitumor activity alone or in combination with anti-MM IMiD [43,80]. Finally, MOR202 and TAK-079 anti-CD38 antibodies are actually in phase I/II clinical trials in relapsed/refractory MM patients (ClinicalTrials.gov Identifier: NCT01421186 and NCT03439280, respectively).

Finally, other strategies, including CD47 targeting [72,73], CD55/CD59 inhibitors [52,83], the enhancement of NK effector functions via ex vivo expanded NK cells [57,67], and the use of bi-specific molecules [57,84], are under investigation and may represent new therapeutic strategies to improve the outcome of Dara-treated MM patients.

## Figures and Tables

**Figure 1 cells-09-00167-f001:**
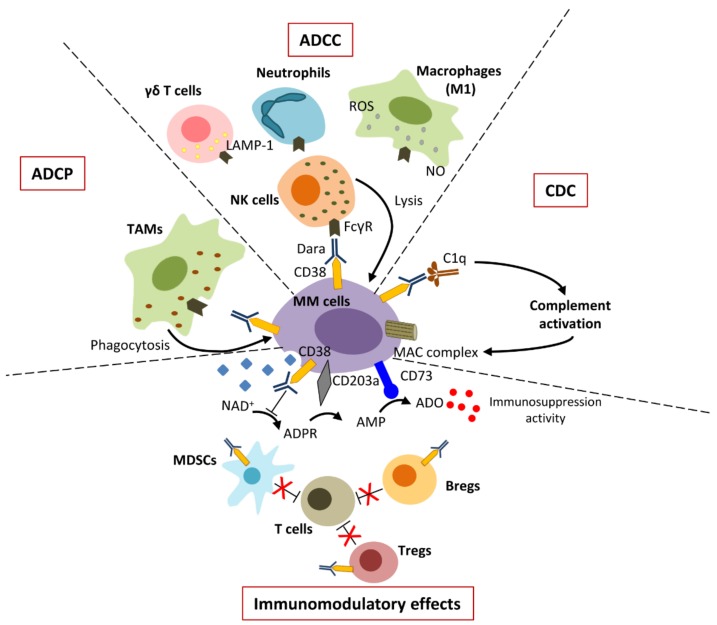
Dara mechanisms of action. Dara recognizes CD38 on MM cells and exerts anti-MM activity via Fc-dependent mechanisms and via immunomodulatory effects. The Fc-dependent mechanisms involve (i) ADCC and (ii) ADCP via the engagement of Dara Fc fragment to FcRs-expressing effector cells (i.e., NK cells, γδ T cells, neutrophils, and macrophages), causing the lysis and/or the phagocytosis of MM cells, respectively; (iii) CDC via the engagement of C1q that activates the complement cascade resulting in the assembly of MAC complex that enables the lysis of the target cells. Dara has also an immunomodulatory effect via inhibition of CD38 ectoenzymatic activity, resulting in a reduction of the immunosuppressive ADO and via the elimination of CD38^+^ immunosuppressive cells (i.e., MDSCs, Tregs, and Bregs): these mechanisms promote T-cell proliferation and effector functions.

**Figure 2 cells-09-00167-f002:**
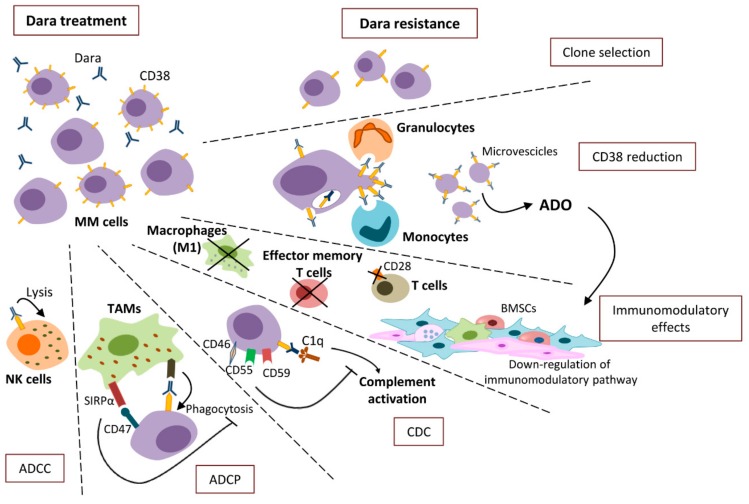
Dara mechanisms of resistance include the following: (i) clone selection of CD38^dim^ MM cells; (ii) CD38 reduction via CD38 endocytosis, trogocytosis by granulocytes and monocytes, and via release of CD38-expressing microvescicles that contribute to ADO production and immunosuppression; (iii) immunomodulatory effects via downregulation of intracellular pathways in BMSCs, a decrease of effector memory T cells, M1 macrophages, and of the co-stimulatory CD28 expression on T cells; (iv) MM cells overexpression of CD46 and of the membrane-associated complement-inhibitory proteins (CD55 and CD59) that prevent CDC; (v) MM cells’ overexpression of CD47 that recognizes SIRPα on TAMs inhibiting ADCP; and (vi) depletion of CD38^+^ NK cells via fratricide ADCC.

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
