# Peer review of "Mechanisms of Resistance to Anti-CD38 Daratumumab in Multiple Myeloma"

_cells, 2020, doi:10.3390/cells9010167_

Round 1
Reviewer 1 Report
The manuscript by Santarella et al. deals with a topic of great interest; however, it present considerable weaknesses and requires major revision both in scientific content and stile.
The authors present a review summarizing the current literature on the Daratumumab-mediated mechanisms of resistance in multiple myeloma. This topic is of considerable interest since Daratumumab became the backbone of the current standards of therapy both in USA and Europe. However, the manuscript is largely out of focus, indeed, the first five pages of the review describe the main mechanisms of action of all therapeutic antibodies with some reference to Dara. I feel that this part should be substantially shortened and the overall structure of the review could be much improved. Moreover, the manuscript is hard to read because of bad English. Here my suggestions and recommendations.
Abstract
Line 17 – 18: the concept can be understood, but the sentence must be rephrased
line 20 - the immunomodulatory effects exerted by Dara are not limited to removal of CD38-positive immune suppressor cells
Introduction
Line 44 – 45: This sentence is inaccurate. Indeed, NAD may regulate many cellular functions, among these, its involvement in local immunosuppression is not its main function. CD38 catalyses cleavage of NAD into ADP-ribose or cyclic ADP-ribose and nicotinamide, as well as the transglycosidation of NADP and nicotinic acid to yield NAADP. Cyclic ADP-ribose and NAADP are second messenger molecules which trigger calcium release from IP3-independent intracellular stores. CD38 may also be involved in the regulation of immune functions by limiting the substrate availability for ADP-ribosyltransferases.
Line 46 – 49: The description of the contribution of CD38 to the generation of immunosuppressive local environment is quite confuse.
Line 56 - what is the meaning of “active patients”?
2.1 ADCC
This paragraph largely refers to the main effector mechanisms of the ADCC induced by therapeutic antibodies that do not necessarily represent documented Dara-mediated effects. For example, macrophages, neutrophils and gamma/delta T cells have been shown to induce ADCC against tumor cells coated with some therapeutic antibody, but their role in Dara-induced ADCC is currently unknown.
2.2 ADCP
The focus of this paragraph (and paragraph 2.3) should be the ability of Dara to promote ADCP (or CDC).
Line 158. This sentence is misleading, I suggest to remove it.
2.4 Immunomodulatory effect (I suggest to use the plural “effecs”)
Line 158: the canonical adenosinergic pathway responsible for the production of ADO in the MM microenvironment is mediated by CD39 and CD73 that convert ATP to ADO. This pathway is flanked by another set of ectoenzymes (including CD38) that leads to ADO production using NAD+ as a starting substrate. This sophisticated mechanism requires an accurate description so as not to cause confusion and misunderstanding.
Line 251. ADCC resistance
The potential impact of BM stroma-MM cell interactions on the ADCC response to daratumumab should be mentioned as microenvironment-related resistance mechanism (Haematologica 2016 Aug;101(8): e339-42).
Line 299. This paragraph is difficult to read and does not allow to understand what are the mechanistic bases of the immuno-mediated Dara resistance.
Line 312. It is not clear how “deregulation of (inhibitory) immune checkpoints” may have a role in the development of resistance to Dara. Actually, the inhibitory immune checkpoints (such as, PD1 and PDL1) are overexpressed in patients, this upregulation, besides being involved in the resistance to PD1 and PDL1 specific inhibitors, may contribute to the development of resistance to the immunomodulatory activities of Dara.
Minor points
Please, remove abbreviations that are not necessary
Legend to figure 1: point iii) referred to CDC, is not correctly described
Legend to figure 2: point iii) referred to Immunomodulatory effect is not clearly described.
Author Response
POINT-BY-POINT ANSWERS TO REVIEWERS COMMENTS
The authors thank the Reviewer 1 for helpful criticism and are glad for positive comments.
Abstract
Reviewer’s comment: Line 17 - 18: the concept can be understood, but the sentence must be rephrased
Reply: We thank the Reviewer for this comment. Accordingly, the sentence has been rephrased as follows: “Although recent data have demonstrated very promising results in clinical practice and trials, some patients do not achieve a partial response, and ultimately all patients undergo progression.”, (page 1, lines 17-18).
Reviewer’s comment: line 20 - the immunomodulatory effects exerted by Dara are not limited to removal of CD38-positive immune suppressor cells.
Reply: According to the Reviewer’s comment the sentence “through removal of CD38+ immune suppressor cells” has been removed (page 1, line 20).
Introduction
Reviewer’s comments: Line 44 - 45: This sentence is inaccurate. Indeed, NAD may regulate many cellular functions, among these, its involvement in local immunosuppression is not its main function. CD38 catalyses cleavage of NAD into ADP-ribose or cyclic ADP-ribose and nicotinamide, as well as the transglycosidation of NADP and nicotinic acid to yield NAADP. Cyclic ADP-ribose and NAADP are second messenger molecules which trigger calcium release from IP3-independent intracellular stores. CD38 may also be involved in the regulation of immune functions by limiting the substrate availability for ADP-ribosyltransferases.
Line 46 – 49: The description of the contribution of CD38 to the generation of immunosuppressive local environment is quite confuse.
Reply: As requested, in the “Introduction” section we modified the sentences about the involvement of CD38 into NAD cleavage as follow: “CD38 contributes to the catabolism of nicotinamide adenine dinucleotide (NAD+) involved in several cellular functions including local immunosuppression [13]. In the BM niche, MM cells reprogram their metabolism that increases NAD+ levels yielding lactate and H+ [18]. Through its ectoenzymatic activity, CD38 catalyzes the cleavage of NAD+ into cyclic adenosine diphosphate (ADP)-ribose, next to ADP-ribose via hydrolase activity [13]. Furthermore, CD38 catalyzes the transglycosilation of nicotinamide adenine dinucleotide phosphate (NADP) and nicotinic acid to yield nicotinic acid adenine dinucleotide phosphate (NAADP). Cyclic ADP-ribose and NAADP are second messenger molecules which trigger calcium release from inositol trisphosphate (IP3) -independent intracellular stores. ADP-ribose is the substrate for the formation of adenosine (ADO) that is involved in the regulation of MM immune autocrine and paracrine signals [18].” (page 2, lines 45-54).
Reviewer’s comment: Line 56 - what is the meaning of “active patients”?
Reply: Multiple myeloma active patients are considered patients with the presence of CRAB (Calcium - Renal failure - Anemia - Bone lesions) features that have overcome the monoclonal gammopathy of uncertain significance (MGUS) and the smoldering stages. Patients with active myeloma have elevated levels of calcium, the presence of renal failure due to the proteins secreted by the malignant cells, an evident anemia and the bone lesions in the spine and ribs. In the text we have replaced “active patients” with “MM patients” (page 3, line 61 and page 5, line 128).
2.1 ADCC
Reviewer’s comment: This paragraph largely refers to the main effector mechanisms of the ADCC induced by therapeutic antibodies that do not necessarily represent documented Dara-mediated effects. For example, macrophages, neutrophils and gamma/delta T cells have been shown to induce ADCC against tumor cells coated with some therapeutic antibody, but their role in Dara-induced ADCC is currently unknown.
Reply: We thank the Reviewer for this suggestion. Accordingly, we modified this paragraph as follow: “Several studies documented that therapeutic mABs sustain ADCC by other immune cells (Figure 1) [28,36,37]. Neutrophils exert cytotoxic activity through the destruction of plasma membrane, leading to the lysis of target cells [36]. Although macrophages are mainly involved in ADCP (see below), M1 macrophages induce ADCC via the release of nitric oxide (NO) and other oxygen reactive species (ROS) [28]. Finally, γδ T cells contribute to target cell killing via ADCC through a CD16-dependent mechanism that induces γδ T cells degranulation and release of the lysosomal-associated membrane protein-1 (LAMP-1 or CD107a) [37]. However the role of neutrophils and γδ T cells in Dara-induced ADCC is currently unknown.” (pages 4-5, lines 109-116).
2.2 ADCP
Reviewer’s comment: The focus of this paragraph (and paragraph 2.3) should be the ability of Dara to promote ADCP (or CDC).
Reply: We thank the Reviewer for this helpful comment. In line with this suggestion the paragraphs 2.2. and 2.3 have been modified as follow:
“2.2 ADCP
ADCP is an Fc-dependent mechanism that supports antibody-mediated clearance of tumor cells both in vitro and in vivo [38]. Specifically, antibody-opsonized target cells activate the FcγRs expressed on the surface of different immune cells that promote the phagocytic process through the internalization and the consequent lysis of target cancer cells via phagosome acidification [38].
Overdijk et al. [39] studied the ability of Dara to induce tumor-associated macrophage (TAM)-mediated phagocytosis in MM (Figure 1). Flow cytometry and live cell imaging revealed that macrophages rapidly engulf CD38+ MM cell lines in vitro. Authors finely demonstrated that phagocytosis contributes to the anti-tumour activity of Dara in vivo by using immune-deficient SCID-BEIGE mice, lacking B, T and NK cells [39]. Dara-dependent phagocytosis is related to CD38 expression, in that MM cell lines with low CD38 levels are not susceptible to Dara effect. They also explored the role of Dara-dependent phagocytosis in fresh purified MM cells derived from MM patients suggesting that phagocytosis is a potent mechanism that contribute to the therapeutic activity and clinical efficacy of Dara in MM patients [39].
2.3 CDC
The engagement of the Fc fragment to C1q initiates the classical pathway of complement cascade, leading to formation of the membrane attack complex (MAC), a transmembrane channel that enables the osmotic lysis of the target cell (Figure 1) [40]. The activation of the complement cascade induces the generation of the active opsonin C3b that covalently binds to glycoproteins on the target cell surface, and allows cell recognition by phagocytic cells and production of the anaphylatoxins C3a and C5a [40]. These pro-inflammatory chemokines modulate several immune responses promoting leukocyte chemotaxis, phagocytes degranulation and the simultaneous up-regulation of activating FcγRIII and down-regulation of inhibitory FcγRs on macrophages [41].
Dara is the most important inducer of CDC. It was selected from a panel of 42 mAbs based on its ability to produce the complement-mediated lysis of the B lymphoblast Daudi cell line and freshly isolated MM cells alone and in co-culture with BMSCs [35]. Dara induces CDC at low concentration of human serum, proving its efficacy in a complement-defective milieu as the BM of MM patients [35]. In addition Nijhof et al. [42] found that Dara-mediated CDC depends on CD38 expression. Indeed, enforced CD38 expression on MM cells significantly improved CDC activity of Dara [42].” (pages 5-6 lines 117-145).
Because of the removal of the references 39 by Weiskopf et al., 40 by Levin et al., 42 by Ribatti et al., 43 by Mantovani et al. and 45 by Schmidt et al., all the other following references have been accordingly re-numbered.
2.4 Immunomodulatory effect
Reviewer’s comment: Immunomodulatory effect (I suggest to use the plural “effecs”).
Reply: We thank the Reviewer for the suggestion and we replaced “Immunomodulatory effect” with “Immunomodulatory effects” (page 6 line 146).
Reviewer’s comment: Line 158. This sentence is misleading, I suggest to remove it.
Reply: As requested, the sentence “The ectoenzymatic activity of CD38 converts extracellular NAD+ in high levels of ADO that creates an immune suppressive tumor microenvironment [18].” has been removed (page 6, lines 147-148).
Reviewer’s comment: Line 158: the canonical adenosinergic pathway responsible for the production of ADO in the MM microenvironment is mediated by CD39 and CD73 that convert ATP to ADO. This pathway is flanked by another set of ectoenzymes (including CD38) that leads to ADO production using NAD+ as a starting substrate. This sophisticated mechanism requires an accurate description so as not to cause confusion and misunderstanding.
Reply: To better explain the sophisticated mechanism to produce ADO in MM microenvironment, we replaced the misunderstanding sentences with: “The production of ADO in the MM microenvironment is regulated by the “canonical” adenosinergic pathway catalyzed by CD39/CD73, and the “alternative” axis catalyzed by CD38/CD203a/CD73 [43,44]. CD38 starts the “alternative” axis converting NAD+ to cyclic ADP-ribose, further metabolized by CD203a into AMP, which is subsequently converted to ADO by CD73 [18, 44]” (page 6, lines 149-153).
Because the reference 43 by Morandi et al. and 44 by Horenstein et al. have been inserted, all the other following references have been re-numbered accordingly.
ADCC resistance
Reviewer’s comment: The potential impact of BM stroma-MM cell interactions on the ADCC response to daratumumab should be mentioned as microenvironment-related resistance mechanism (Haematologica 2016 Aug;101(8): e339-42).
Reply: We thank the Reviewer for this suggestion. Accordingly, we have added the role of BM microenvironment in supporting Dara resistance to ADCC as follow: “Finally, de Haart et al. [68] showed that BMSCs support resistance to Dara therapy preventing Dara-mediated ADCC. The microenvironment-related resistance is mediated by the overexpression of the anti-apoptotic protein survivin in MM cells upon interaction with BMSCs. Indeed, treatment of MM cells/BMSCs co-cultures with YM155, a small molecule that inhibits survivin expression, increased Dara-mediated ADCC abrogating the protective role of BM microenvironment against Dara treatment [68].” (page 9, lines 253-258).
Because the reference 68 by de Haart et al. has been inserted, all the other following references have been accordingly re-numbered.
Reviewer’s comment: Line 299. This paragraph is difficult to read and does not allow to understand what are the mechanistic bases of the immuno-mediated Dara resistance.
Reply: We thank the Reviewer for this helpful comment. The paragraph bases on few observations showing a low number of effector T cells in non-responder Dara-treated patients. In addition, the mentioned results of Viola et al. and Neri et al. have been described in an abstract without a description of the mechanistic basis of the the immuno-mediated Dara resistance. According to Reviewer’s suggestion, we have modified the paragraph 3.5 as follow:
“3.5 The immune-mediated resistance
Recently, some Authors [49,78,79] hypothesize that immune system may play a key role in the development of Dara resistance.
Krejick et al. [49] and Neri et al. [78] documented a low numbers of effector memory T cells in relapsed MM patients. Single cell resolution analysis of BMMCs depleted of CD138+ cells from Dara/IMiD-treated patients showed a decreased expression of the co-stimulatory antigen CD28 on T cells and a reduced number of M1 macrophages in resistant and/or progressing patients compared to responder ones [78]. Finally, RNA-sequencing of BMSCs cells depleted of CD138+ MM cells showed a different gene expression profile between progressed and Dara naïve patients. The Database for Annotation, Visualization and Integrated Discovery (DAVID) analysis provides the functional interpretation of the down-regulated genes, such as tool like receptor (TLR) 8, CD47, chemokine (C-X-C motif) ligand (CXCL) 10 and CXCL4, highlighting their involvement in the Dara-mediated immune-response (Figure 2) [79].
Nevertheless, this intriguing hypothesis should be further investigated. ” (pages 10-11, lines 288-301).
Because the reference 78 by Neri et al. has been inserted, all the other following references have been accordingly re-numbered.
Reviewer’s comment: Line 312. It is not clear how “deregulation of (inhibitory) immune checkpoints” may have a role in the development of resistance to Dara. Actually, the inhibitory immune checkpoints (such as, PD1 and PDL1) are overexpressed in patients, this upregulation, besides being involved in the resistance to PD1 and PDL1 specific inhibitors, may contribute to the development of resistance to the immunomodulatory activities of Dara.
Reply: We are grateful for the helpful comment of the Reviewer. We agree with the observation that this part is misleading and confusing. Although preclinical studies proved the synergistic effect of anti-CD38 antibodies and PD-1/PD-L1 inhibitors against different solid and haematological tumors [Chen et al., Cancer Discovery, 2018; Koyama et al. Nat Commun, 2016; Syn et al. Lancet Oncol, 2017], the involvement of PD-1/PD-L1 axis in the development of resistance to Dara therapy need to be further investigated. Nevertheless, a cross-talk between CD38 and PD1/PDL1 axis has been documented [Chen et al., Cancer Discovery, 2018; ]. For these reasons, we have moved this part in the paragraph 2.4 as follow: “Chen et al. [50] demonstrated that overexpression of CD38 is a mechanism of tumor escape from PD1/PDL1 axis blockade in that it suppressed CD8+ T cell function via ADO signaling. Consistent with these results, Bezman et al. [51] showed an enhanced anti-tumor activity in MC38 and J558 mice treated with a combined therapy suggesting that a dual targeting of CD38 and PD1 may represent a promising anti-MM therapeutic strategy. The synergic effect induced by anti-CD38 mAbs was related to a reduction in the frequency of immunosuppressive Tregs and MDSCs populations.” (page 7, lines 175-180).
Because the removal of the reference 79 by Koyama et al., 80 by Syn et al. and the addition of the new reference 51 by Bezman et al., all the other following references have been accordingly re-numbered.
According to text adjustment, Figure 2 and Legend to Figure 2 have been modified as follow:
Figure 2. Dara mechanisms of resistance. These are: i) Clone selection of CD38dim MM cells; ii) CD38 reduction via CD38 endocytosis, trogocytosis by granulocytes and monocytes and via release of CD38-expressing microvescicles that contribute to ADO production and immunosuppression; iii) immunomodulatory effects via down-regulation of intracellular pathways in BMSCs, a decrease of effector memory T cells, M1 macrophages and of the co-stimulatory CD28 expression on T cells; iv) MM cells overexpression of CD46 and of the membrane-associated complement-inhibitory proteins (CD55 and CD59) that prevent CDC; v) MM cells overexpression of CD47 that recognizes SIRPα on TAMs inhibiting ADCP; vi) depletion of CD38+ NK cells via fratricide ADCC.”.
Minor points
Reviewer’s comment: Please, remove abbreviations that are not necessary
Reply: According to reviewer suggestion we have removed some not necessary abbraviations, i.e. overall survival (OS) (page 2, line 32), complete response (CR) (page 3, line 71). We left the other abbreviations since they are mentioned more than once in the text and/or in the figures.
Reviewer’s comment: Legend to figure 1: point iii) referred to CDC, is not correctly described.
Reply: According to Reviewer 1, legend to Figure 1 has been modified as follow: “Figure 1. Dara mechanisms of action. Dara recognizes CD38 on MM cells and exerts anti-MM activity via Fc-dependent mechanisms and via immunomodulatory effects. The Fc-dependent mechanisms involve: i) ADCC and ii) ADCP via the engagement of Dara Fc fragment to FcRs-expressing effector cells (i.e. NK cells, γδ T cells, neutrophils, macrophages), causing the lysis and/or the phagocytosis of MM cells, respectively; iii) CDC via the engagement of C1q that activates the complement cascade resulting in the assembly of MAC complex that enables the lysis of the target cells. Dara has also an immunomodulatory effect via inhibition of CD38 ectoenzymatic activity resulting in a reduction of the immunosuppressive ADO and via the elimination of CD38+ immunosuppressive cells (i.e. MDSCs, Tregs, and Bregs): these mechanisms promote T cell proliferation and effector functions.”.
Reviewer’s comment: Legend to figure 2: point iii) referred to Immunomodulatory effect is not clearly described.
Reply: According to Reviewer suggestion and to text adjustment, legend to Figure 2 has been modified as follow: “Figure 2. Dara mechanisms of resistance. These are: i) Clone selection of CD38dim MM cells; ii) CD38 reduction via CD38 endocytosis, trogocytosis by granulocytes and monocytes and via release of CD38-expressing microvescicles that contribute to ADO production and immunosuppression; iii) immunomodulatory effects via down-regulation of intracellular pathways in BMSCs, a decrease of effector memory T cells, M1 macrophages and of the co-stimulatory CD28 expression on T cells; iv) MM cells overexpression of CD46 and of the membrane-associated complement-inhibitory proteins (CD55 and CD59) that prevent CDC; v) MM cells overexpression of CD47 that recognizes SIRPα on TAMs inhibiting ADCP; vi) depletion of CD38+ NK cells via fratricide ADCC.”.
Reviewer 2 Report
This is a well written and niche filling manuscript that deals with resistance to the registered anti CD38 molecule, daratumumab. The paper is easily digestable for the readers, complex data are well presented. In this review, the authors sytematically introduce the multiple anti-myeloma mechanisms of action of this antibody. Thereafter, details of resistance mechanisms to DARA are presented. Last, but not least, recent approaches to overcome drug resistance are discussed.
I only missed mentioning the differences between daratumumab and isatuximab (and possibly the alternative MOR and TAK anti-CD38 antibodies), though admittedly there are much fewer data available. Maybe it could be added to the discussion section.
Citation of the recent Cells paper by Morandi and coworkers is also recommended.
Author Response
POINT-BY-POINT ANSWERS TO REVIEWERS COMMENTS
The authors thank the Reviewer 2 for helpful criticism and are glad for positive comments.
Reviewer’s comment: I only missed mentioning the differences between daratumumab and isatuximab (and possibly the alternative MOR and TAK anti-CD38 antibodies), though admittedly there are much fewer data available. Maybe it could be added to the discussion section. Citation of the recent Cells paper by Morandi and coworkers is also recommended.
Reply: We thank the Reviewer for this suggestion. Accordingly, we added the new reference 43 by Morandi et al. and a description of the other anti-CD38 antibodies in the section “New strategies to overcome Dara resistance”, as follow: “Another strategy to overcome resistance to Dara may be the use of other anti-CD38 antibodies with a different mechanisms of action (van de donk mechanism of action/resistance), namely Isatuximab (SAR650984), MOR202 and TAK-079 [80]. Isatuximab mediates a direct cytotoxicity against MM cells in addition to the canonical Fc-dependent mechanisms of action [81]. Indeed, Jiang et al. demonstrated that it induces a CD38-dependent depletion of MM cells via homotypic aggregation-associated cell death by actin cytoskeleton polymerization, caspase-dependent apoptosis and lysosomal cell death [81] Furthermore, Isatuximab induces an allosteric modulation of CD38 that results in a higher inhibition of its ecto-enzymatic activity [82]. Clinical trials demonstrated that Isatuximab has a great anti-tumor activity alone or in combination with anti-MM immunomodulatory drugs [43,80]. Finally, MOR202 and TAK-079 anti-CD38 antibodies are actually in phase I/II clinical trial in relapsed/refractory MM patients (ClinicalTrials.gov Identifier: NCT01421186 and NCT03439280, respectively).” (pages 11-12, line 318-328).
Because the references 80 by Frerichs, 81 by Jiang et al. and 82 by Deckert et al. have been inserted, all the other following references have been re-numbered accordingly.
Reviewer 3 Report
The review by Saltarella et al. provides an interesting and complete overview of the current knowledge of the the mechanisms involved in Daratumumab efficacy and resistance, focusing on both tumor cells and microenvironment. The manuscript gives a nice introduction of the mechanism of action of the anti-CD38 mAb, and in a second part of the several mechanisms that contribute to the development of resistance. The review is nicely written, and the references have been cited in context and on topic. The review is precisely prepared thus, only a minor change would be recommended:
1- Line 279: The reference to paper 73 by Jaiswal and colleagues is ok but the conclusion from this paper is lacking and misleading (only contribute to the finding that overexpression of CD47 on leukemia cells increases its pathogenicity by allowing it to evade phagocytosis).
Author Response
POINT-BY-POINT ANSWERS TO REVIEWERS COMMENTS
The authors thank the Reviewer 3 for helpful criticism and are glad for positive comments.
Reviewer’s minor comment: 1- Line 279: The reference to paper 73 by Jaiswal and colleagues is ok but the conclusion from this paper is lacking and misleading (only contribute to the finding that overexpression of CD47 on leukemia cells increases its pathogenicity by allowing it to evade phagocytosis).
Reply: We thank the Reviewer for this helpful comment. In the “ADCP resistance” section, we removed the reference 73 by Jaiswal et al. that focus only on CD47 in leukemia cells and we modified the paragraph 3.3 as follow:
“3.3 ADCP resistance
Another important mechanism of Dara resistance involves CD47, also known as integrin-associated protein (IAP), an antigen identified in different tumors, including MM [69]. CD47 is expressed at low levels on normal plasma cells, while it is overexpressed on MM cells [70]. CD47 over-expression may contribute to the immune escape of tumor cells through the inhibition of ADCP, via the binding of CD47 to the signal-regulatory protein alpha (SIRPα) on TAMs [71]. The CD47/SIRPα complex acts as a “don't eat me” signal that induces SIRPα phosphorylation and association to Src-homology phosphatase 1 domain (SHP-1) on macrophages resulting into the inhibition of phagocytosis (Figure 2) [73]. The “don't eat me” signal is the focus for different mAb therapies able to block CD47 [72,73], indeed van Bommel et al. [73] demonstrated an increase of ADCP through CD47-blocking mAbs and the consequent regulation of the anti-phagocytic activity. Specifically, they described the RTX-CD47 bi-specific antibody that recognizes CD47 single chain fragment in the antibody variable regions (scFv) and in tandem CD20-targeting scFv derived from the rituximab association [73]. RTX-CD47 promotes the inactivation of CD47/SIRPα signal and causes the selective removal of CD47+CD20+ cells through phagocytosis. Overall suggests that the block of CD47 may act as a pro-phagocytic therapeutic approach to enhance the tumoricidal activity of anticancer mAbs, including Dara [73].” (pages 9-10, lines 259-274).
Because the removal of the reference 73 by Jaiswal et al., all the other following references have been accordingly re-numbered.
Reviewer 4 Report
minor
page 2, line 61: single agent in relapsed/refractory MM as well as for patients with at least one line of treatment and in combination with either DRd or DVd
page 2, line 67 add: In transplant eligible patients, Cassiopeia study (Lancet 2019)....
page 12, line 487: J Clin Oncol
Ref 54 is incomplete
Author Response
POINT-BY-POINT ANSWERS TO REVIEWERS COMMENTS
The authors thank the Reviewer 4 for helpful criticism and are glad for positive comments.
Reviewer’s minor comment: page 2, line 61: single agent in relapsed/refractory MM as well as for patients with at least one line of treatment and in combination with either DRd or DVd.
Reply: According to Reviewer’s comment, the text has been changed: “Due to these results, Dara is currently approved by the European Medicines Agency (EMA) and also by the U.S. Food and Drug Administration (US FDA) as a single agent in relapsed/refractory MM as well as for patients with at least one line of treatment and in combination with either Dara-Revlimid-Dexamethasone (DRd) and Dara-Velcade-Dexamethasone (DVd) in patients with at least one previous line of therapy.” (page 3, lines 66-67).
Reviewer’s minor comment: page 2, line 67 add: In transplant eligible patients, Cassiopeia study (Lancet 2019)....
Reply: We are grateful for this suggestion. We have therefore added a new reference 29 and a new sentence: “In transplant-eligible patients with newly diagnosed MM, CASSIOPEA study [26] demonstrated that the association of Dara with standard of care Velcade-Thalidomide-Dexamethasone (VTD) improves depth of response and PFS suggesting a concrete change in the scenario of this stage of disease.” (page 3, lines 73-76).
Because the reference 26 by Moreau et al. has been removed, all the other following references have been re-numbered accordingly.
Reviewer’s minor comment: page 12, line 487: J Clin Oncol.
Reply: As requested, the new reference 53 has been corrected: “Boxhammer, R.; Steidl, S.; Endell, J. Effect of IMiD compounds on CD38 expression on multiple myeloma cells: MOR202, a human CD38 antibody in combination with pomalidomide. J of Clin Oncol. 2015, 33, 8588, [abstract].”
Reviewer’s minor comment: Ref 54 is incomplete.
Reply: The new reference 54 has been corrected: “Endell, J.; Boxhammer, R.; Wurzenberger, C.; Ness, D.; Steidl, S. The Activity of MOR202, a Fully Human Anti-CD38 Antibody, Is Complemented by ADCP and Is Synergistically Enhanced by Lenalidomide in Vitro and in Vivo. Blood 2012, 120, 4018, [abstract].”
Round 2
Reviewer 1 Report
The revised version of the manuscript has been significantly improved and now warrants publication in Cells.